# Smartphone use patterns and problematic smartphone use among preschool children

**Jeong Hye Park**[1]ORCID[☯]*, **Minjung Park**[2]ORCID[☯]

**1** Department of Nursing, Gyeongsang National University, Jinju-si, Gyeongsangnam-do, Republic of Korea,
**2** National Agency for Development of Innovative Technologies in Korean Medicine, National Institute of Korean Medicine Development, Seoul, Republic of Korea

☯ These authors contributed equally to this work.
* masternur@naver.com

## Abstract

### Background

The purpose of the present study was to identify smartphone use patterns associated with problematic smartphone use (PSU) among preschool children. Little is known about PSU patterns in younger children, although the age for first smartphone use is decreasing.

### Methods

We applied a cross-sectional study design to analyze data obtained from a nationwide survey on smartphone overdependence conducted in 2017 by the South Korean Ministry of Science and ICT and the National Information Society Agency. Data from 1,378 preschool children were analyzed using binomial logistic regression analysis. This study was conducted in compliance with STROBE (Strengthening the Reporting of Observational Studies in Epidemiology).

### Results

Seventeen percent of the sample met the criteria for PSU. The odds of PSU significantly increased with frequent smartphone use and in children who used a smartphone for more than two hours per day. Using smartphones to watch TV shows or videos for entertainment or fun significantly increased the odds of PSU, whereas using smartphones for education, games, and social networking did not.

### Conclusions

The findings indicate that one of five preschool children using smartphones could experience PSU. Compared to other age groups, PSU in young children may be more associated with their caregivers. To prevent PSU in preschool children, caregivers need information about the total screen time recommended for children, smartphone use patterns associated with PSU, suggestions for other activities as possible alternatives to smartphone use, and strategies to strengthen children's self-regulation with regards to smartphone use.

**Data Availability Statement:** The datasets analyzed during the current study are available in the Ministry of Science and ICT and the National Information Society Agency [https://www.data.go.kr/en/index.do]. Data access requests can be

submitted through the 'Request Data' function in the website menu.

**Funding:** This study was supported by the Korean Medicine R&D program funded by Ministry of Health & Welfare through the Korea Health Industry Development Institute (KHIDI) (HI20C0867). The funder had no role in study design, data collection and analysis, decision to publish, or preparation of the manuscript.

**Competing interests:** The authors have declared that no competing interests exist.

## Background

By 2021, the population of smartphone users worldwide is expected to reach 3.8 billion, and to continue its rapid growth [1]. A smartphone is a mobile phone that can be used as a handheld computer and connects to the internet [2]. Smartphone applications (apps) are computer programs designed for a specific purpose [3] and are widely developed and used. Convenient internet access using apps has resulted in smartphones rapidly becoming a necessity in everyday life; smartphones help people work, study, acquire or share information, create or maintain social relationships, and enjoy leisure activities.

Despite these benefits, however, a growing body of literature reports negative consequences and possible dangers associated with smartphones [4–7]. Of the negative effects of smartphone use, problematic smartphone use (PSU) involves excessive or uncontrollable use, preoccupation, neglect of other activities, and continued use despite the evidence of potential harm [4–6]. Moreover, PSU may have various negative effects on children's physical and psychological health, social interaction, and cognitive development; excessive exposure to smartphones has been associated with poor vision, poor sleep quality, and psychological dependence [4–6]. Children may be disconnected from actual social interactions, including interactions with other children and the people around them. They may also have difficulties in deep thinking and concentration since many smartphone apps are designed to be accessed and used intuitively, and to easily allow moves from one option to another to get interesting information. This could harm brain development and educational attainment [4–6].

Factors associated with PSU in middle and late childhood have been found to be the excessive use of entertainment apps (e.g., games), instant messengers, and social networking services (SNS) [5, 6, 8–11], as well as high frequency and long duration smartphone use patterns [6]. Furthermore, children's PSU has been seen to contribute to a low quality of life [12], low quality of friendship [13], and poor mental health outcomes [5].

Recently, studies show that, on average, smartphones are first used at age four to five [14–16], and the age of first exposure is gradually decreasing [17–19]. Preschool children, who are between three to six years old, are making significant progress in cognitive, linguistic, and psychosocial development [20]. Inappropriate smartphone use patterns formed in early childhood can have a serious impact on a child's future.

Previous studies report that the average cumulative screen time of preschool children was 4.1 hours per day, which is beyond the recommended amount of time [21], and preschoolers' excessive screen time could increase their risk of inattention problems [22]. However, the effect of excessive smartphone use in preschool children has not yet been well studied.

South Korea has a high penetration rate of smartphone and internet use. In 2017, 89.5% of people over age three had smartphones [23] and 90.3% used the internet [24]. In addition, 99.4% of internet users have accessed the internet wirelessly, most using a smartphone (94.1%) [24]. Among smartphone users in South Korea, the prevalence of PSU among children under age 10 rose from 17.9% in 2016 to 20.7% in 2018, the largest increase among all age groups [23]. Even for young children, smartphones are more accessible and universal than other devices [14–19]. Therefore, it is important to identify smartphone use patterns and sociodemographic characteristics contributing to PSU in preschool children. Understanding PSU and associated factors in preschool children is critical for developing appropriate prevention strategies.

## Methods

### Design

We applied a cross-sectional study design that complied with the Strengthening the Reporting of Observational Studies in Epidemiology (STROBE) statement [25].

## Participants and data collection

**Participants.**  Data were collected by asking children's main caregivers in 1,279 households to observe and report on their child's behaviors or attitudes. The data from 1,378 preschool children aged three to six who attended daycare centers or kindergartens were analyzed. The main caregiver was defined as the person who plays the primary role in caring for the child; this person was asked to report on the child's activities at home.

**Data collection.**  All data in the present study were obtained from the 2017 South Korean nationwide Survey on Smartphone Overdependence conducted by the Ministry of Science and ICT and the National Information Society Agency [26]. This survey is conducted annually to secure data for policymaking on appropriate smartphone use. The 2017 survey targeted smartphone users age 3–69 in all households nationwide as of September 1, 2017. After stratifying the population into 17 cities/provinces, the enumeration district was derived as the first sampling unit using proportional probability to size the systematic sampling in proportion to the number of households. The number of households was derived as the second sampling unit using systematic sampling. The complete enumeration of smartphone users age 3–69 was conducted in 10,000 households nationwide. The number of participants was 29,712. Smartphone users were defined as individuals who used the internet access feature of smartphones at least once a month, which aligned with recent changes in defining internet use since internet features became mobile and more accessible [23, 24].

Data were collected via interview surveys by trained investigators who visited participants' households between September and November 2017. The collected data were verified first by the supervisor of a given enumeration district; a second verification was conducted via telephone with a random sample of over 30% of the collected questionnaires. As a third verification, statistics experts checked the range of responses, the divergence between questions, and the consistency of responses. If there were any errors in the data, the participant was contacted to collect the ancillary data, and the data were supplemented or re-examined to verify their validity.

## Measurements

**Smartphone use patterns.**  Smartphone use patterns included frequency and duration per day, the type of the accessed app, and the degree of app use. The average frequency and duration of smartphone use were collected by examining global estimates of media use [27] to find out how much preschool children tend to use a smartphone on average. The question used to gauge the frequency of use was, "How often did your child use a smartphone on a typical day in the last month?" Question steps were used to derive a more accurate answer. Each participant was requested to search their memory for all episodes of her/his child's smartphone use ranging from one day to several days prior to the questionnaire, to separate unusual activities from smartphone-related activities, and to add up all the smartphone-use-episode lengths across days in the last week and last month sequentially. The duration of use was estimated by multiplying the average time per use by the frequency of use per day. Trained investigators helped participants to relay the smartphone use of their children in realistic terms, rather than according to any distorted self-image. App type was classified based on factors found to be related to PSU in previous studies: web surfing, games, television (TV)/video, music, webtoon/fiction, instant messengers, SNS, and education [5, 6, 8–11]. Games, TV/video, music, and webtoon/fiction were for entertainment or fun in the present study, whereas educational apps were designed around curricula with specific goals to form desirable habits, to communicate academic or social skills, or to teach intended lessons [28] such as handwashing habits, instructive fairy tales, math fundamentals, English as a second language, science, crossword puzzles,

and so on. The degree of use for each app type was measured on a seven-point Likert scale (0 = do not use at all, 1 = use rarely, 7 = use very frequently). Smartphones included tablet devices.

**Problematic smartphone use.**   Problematic smartphone use for preschool children was measured with the Korean-language Smartphone Overdependence Scale (S-scale) for children [26, 29]. The S-scale was developed to screen for PSU in each age group, including children, adolescents, adults, and elderly people, based on the Smartphone Addiction Proneness Scale (SAPS), developed in 2011 [30] and the SAPS, restructured in 2014 [31]. This scale focuses on the propensity of problematic use rather than addiction criteria [29]. The scale for children is an observer scale and consists of three subscales, including the main concept of PSU: self-control failure, salience, and serious consequences. "Self-control failure" refers to a condition in which a person is not able to control her/himself on smartphone use according to self-set goals. A sample item (reverse-scored) was "my child stops using the smartphone on his/her own without parental involvement at a given time." "Salience" is the degree to which smartphone use becomes the most salient and important activity in one's daily life. A sample item was "my child loves to play with a smartphone more than anything else." "Serious consequences" refers to negative physical, psychological, and social consequences resulting from PSU. A sample item was "my child rarely plays or learns except for on his/her smartphone." The scale contained nine items evaluated on a four-point scale (1 = strongly disagree, 4 = strongly agree), with higher scores indicating more serious PSU. In the scale development study [29], the threshold for PSU was 24 points or higher out of 36 points, which was derived by the proposed criteria of Internet Gaming Disorder in the DSM-5 [32]. The threshold point indicates an early stage of smartphone dependence in which parents and their children are experiencing conflicts in playing and learning with regard to smartphone use [29]. The study sample for the prior research on scale development comprised 201 parents of children under 10 years old [29]. In the receiver operating characteristic curve, the sensitivity of the reference point was 0.70, the specificity was 0.63, and the area under the receiver operator curve was 0.67 (standard error 0.05, p<.001). Cronbach's alpha for the scale at validation was 0.75; Cronbach's alpha in the present study was 0.80.

**Adjusting variables.**   The following sociodemographic characteristics of children and family were included: child's sex, child's age, family size, household income, parents' employment status, the number of family members, main caregiver in the family, and the age of the main caregiver. In addition, the main caregiver's perception of his/her child's smartphone dependence was assessed with the question, "How much do you think your child is dependent on a smartphone compared to the other children around him/her?" with responses rated on a scale from 1 ('not dependent at all') to 5 ('very dependent').

## Ethical considerations

The present study was approved by the Institutional Review Board of the Korea National Institute for Bioethics Policy (IRB number: P01-201905-21-011) for using the data obtained from the Ministry of Science and ICT and the National Information Society Agency [26]. Informed consent was exempted since the data was de-identified for the participants' personal information and provided to researchers.

## Data analysis

All data were analyzed using IBM SPSS statistics, version 23.0 for Windows (IBM Corporation, U.S.). First, data cleaning was performed to bring the quality of data to a reliable level according to practice guidelines [33]. Descriptive statistics (frequencies, percentages, means, and

standard deviations) were used to examine the sex and age of each child, family size, parents' employment status, household income, the number of family members, the main caregiver in the family, and the age of the main caregiver, the perception of the main caregiver on his/her child's smartphone dependence and use patterns. Crude differences between the non-PSU group and the PSU group were analyzed by $\chi^2$ test and independent samples t-test. Binary logistic regression was also employed to investigate the potential risk factors of PSU in pre-school children. We entered all variables listed in Tables 1 and 2 in the logistic regression model because selection bias may affect the significance of the results in univariate regression [34]. Household income was a nominal variable, with the dummy variable categories used in the regression analysis derived from the univariate analysis results presented in Table 1.

### Differences between non-PSU and PSU in smartphone use patterns

Table 2 shows the differences in smartphone use patterns between the two groups. Smartphone use duration was significantly longer in the PSU group ($\chi^2$ = 564.722, p < 0.001); the PSU group spent 2.5 hours per day (SD 1.86), whereas the non-PSU group spent 0.8 hours (SD 0.47) (t = -14.082, p < 0.001). However, the PSU group spent 8.2 minutes (SD 10.72) per use and the non-PSU group spent 13.1 minutes (SD 7.79) (t = 8.176, p< 0.001). Smartphone use frequency was also significantly higher in the PSU group (t = -12.362, p<.001); 28.0 (SD 29.28) per day in the PSU group and 4.4 (SD 3.04) per day in the non-PSU group. The highest degree of use among app types was education and games in the non-PSU group; 3.1 (SD 2.01) and 3.1 (SD 2.38) out of 7 points, respectively. The highest degree of use in the PSU group was games and TV/video watching: 4.2 (SD 2.29) and 4.0 (SD 2.30) respectively. The TV/video (t = -8.004, p<.001) and games (t = -7.083, p<.001) differed the most in the degree of use between the two groups. Education app use did not differ significantly between the two groups (t = 0.552, p = .582).

## Results

### Differences between non-PSU and PSU by children's and family characteristics

Of the 1,378 children, 1,142 (82.9%) were placed in the non-PSU group and 236 (17.1%) in the PSU group. Table 1 shows the differences in the characteristics of the children and families between the two groups. The PSU group was slightly older than the non-PSU group and the difference was significant (t = -2.404, p = .017). Household income distribution was also significantly different between the two groups (t = 60.317, p<.001). In the PSU group, the main caregiver was more often a parent (t = 11.323, p<.001), and their average age was younger (t = 3.351, p<.001). Caregivers in the PSU group had a higher awareness of their children's smartphone dependence (t = -7.024, p<.001), but the average score was 3.0 out of 5 (SD 0.83), which implied that they perceived their child's smartphone use to be similar to that of other children.

### Potential risk factors for PSU

A multivariate binominal logistic regression analysis was performed to investigate the potential risk factors for PSU in preschool children (Table 3). Using a smartphone for more than two hours per day contributed most to the increased odds of being in the PSU group (OR = 7.85, 95% CI = 3.03–20.31). Moreover, higher smartphone use frequency also increased the odds of PSU (OR = 1.39, 95% CI = 1.32–1.48). TV/video watching significantly increased the odds of PSU (OR = 1.17, 95% CI = 1.02–1.35). However, education was negatively associated with PSU

**Table 1. Differences between non-PSU and PSU according to the characteristics of the children and families (N = 1,378).**

| Variables | Categories | Smartphone use | | t/$\chi^2$ | p |
|---|---|---|---|---|---|
| | | Non-PSU (N = 1,142) | PSU (N = 236) | | |
| | | N (%) or mean (SD) | | | |
| **Children' characteristics** | | | | | |
| Sex | Male | 526 (46.1) | 100 (42.4) | 1.072 | .315 |
| | Female | 616 (53.9) | 136 (57.6) | | |
| Age (years) | | 4.6 (1.11) | 4.8 (1.06) | -2.404 | .017 |
| **Family characteristics** | | | | | |
| Family size (number of members) | | 3.6 (0.92) | 3.5 (0.81) | 1.333 | .183 |
| Parents' employment status | Both | 452 (39.6) | 99 (41.9) | 0.458 | .512 |
| | One | 690 (60.4) | 137 (58.1) | | |
| Household income* Per month (1,000 KRW) | Less than 2,000 | 34 (3.0) | 6 (2.5) | 60.317 | <.001 |
| | 2,000–4,000 | 419 (36.7) | 127 (53.8) | | |
| | 4,000–6,000 | 271 (23.7) | 76 (32.2) | | |
| | 6,000–8,000 | 149 (13.0) | 12 (5.1) | | |
| | 8,000–10,000 | 92 (8.1) | 9 (3.8) | | |
| | More than 10,000 | 177 (15.5) | 6 (2.5) | | |
| Number of preschool children in the household | 1 | 977 (85.6) | 204 (86.4) | 0.692 | .707 |
| | 2 | 162 (14.2) | 32 (13.6) | | |
| | 3 | 3 (0.3) | 0 (0) | | |
| Main caregiver | Parent | 939 (82.2) | 215 (91.1) | 11.323 | <.001 |
| | Grandparent | 203 (17.8) | 21 (8.9) | | |
| Main caregiver's age (years) | | 40.8 (9.83) | 38.9 (7.61) | 3.351 | .001 |
| Main caregiver's perception of his/her child's smartphone dependence | | 2.5 (0.83) | 3.0 (0.83) | -7.024 | <.001 |

Abbreviation: PSU = problematic smartphone use, SD = standard deviation

*One million KRW was 915 USD and mean household income was 4,300 thousand KRW in Korea in 2017.

(OR = 0.84, 95% CI = 0.71–0.98). Both parents being employed significantly increased the PSU odds ratio (OR = 2.03, 95% CI = 1.05–3.91). Lower household income also significantly increased the PSU odds ratio (OR = 11.24, 95% CI = 3.56–35.48; OR = 9.32, 95% CI = 3.07–28.26, respectively). The main caregiver's age was also positively associated with PSU (OR = 1.06, 95% CI = 1.00–1.12).

## Discussion

This study examined the prevalence and potential risk factors of PSU in preschool children, which have not yet been explored in detail. We found PSU prevalence among preschool children to be 17.1%, while previous studies have reported a PSU prevalence of 23.3% on average in middle and late childhood, adolescence, and young adulthood, ranging between 10 and 30% [5]. Although lower than that seen in older children, PSU prevalence in preschool children is not low. Moreover, many children as young as three to four years old can use mobile devices and participate in media multitasking without assistance [17]. The PSU rates among children are increasing every year [23, 24], making it a prominent issue that raises concerns about the potential impact on society and the need for countermeasures.

Among the potential risk factors for PSU, smartphone use frequency increased the odds of being in the PSU group by 1.39 times. The PSU group's mean smartphone use frequency (28.0 times per day) was about six times higher than that of the non-PSU group (4.4 times per day).

**Table 2. Differences between non-PSU and PSU in smartphone use patterns (N = 1,378).**

| Variables | Categories | Smartphone use | | t/$\chi^2$ | p |
|---|---|---|---|---|---|
| | | Non-PSU (N = 1,142) | PSU (N = 236) | | |
| | | N (%) or mean (SD) | | | |
| Frequency of use (times per day) | | 4.4 (3.04) | 28.0 (29.28) | -12.362 | <.001 |
| Duration of use (hours per day) (minutes per use) | | 0.8 (0.47) | 2.5 (1.86) | -14.082 | <.001 |
| | Less than 0.5 | 223 (19.5) | 15 (6.4) | 564.722 | <.001 |
| | 0.5–1 | 737 (65.5) | 37 (15.7) | | |
| | 1–1.5 | 115 (10.1) | 46 (19.5) | | |
| | 1.5–2 | 49 (4.3) | 28 (11.9) | | |
| | More than 2 | 18 (1.6) | 110 (46.6) | | |
| | | 13.1 (7.79) | 8.2 (10.72) | 8.176 | <.001 |
| Degree of use | Education | 3.1 (2.01) | 3.0 (2.62) | 0.552 | .582 |
| | Web surfing | 1.4 (1.98) | 0.9 (1.87) | 3.242 | .001 |
| | Games | 3.1 (2.38) | 4.2 (2.29) | -7.083 | <.001 |
| | TV/video | 2.6 (2.54) | 4.0 (2.30) | -8.004 | <.001 |
| | Music | 2.6 (2.46) | 2.7 (2.50) | -0.498 | .619 |
| | Webtoons/fiction | 0.6 (1.56) | 1.1 (2.19) | -3.128 | .002 |
| | Messenger | 1.4 (2.09) | 1.4 (2.27) | 0.164 | .869 |
| | SNS | 0.7 (1.65) | 0.6 (1.44) | 1.269 | .205 |

Abbreviation: PSU = problematic smartphone use, SD = standard deviation, SNS = social networking services.

Conversely, children in the PSU group used smartphones for an average of 8.2 minutes in a single use, while children in the non-PSU group used them for 13 minutes. Consequently, the children in the PSU group used smartphones more often but for shorter periods than those in non-PSU group, which implies that the desire to use smartphones arises repeatedly and is very easily satisfied. This finding suggests that frequent smartphone use could be an indicator of PSU [6, 11].

Among the app types, TV/video apps significantly increased the odds of PSU, whereas game, messenger, and SNS apps were not related to PSU. Education was significantly negatively associated with PSU, which is consistent with previous study results [10, 35]. The results show that the main concern for preschool children with PSU is TV/video app use. Watching TV/videos is associated with increased screen time and sedentary behavior [36–39]. Screen time includes the use of any electronic screens, such as TV/video/DVD, computers, smartphones, tablet devices, etc. [36]. Longer screen time could be harmful to various aspects of children's health and development [38–40], and may reduce exposure to family, social, and physical activities [41]. Until recently, the primary screen time medium was TV [38, 39]. However, recent trends suggest that the primary screen medium may be switching from TV to smartphones because smartphones are portable and easily accessible from anywhere. Smartphones can lead to longer screen time and hence have a more serious influence on children than TV. Preschool children spend much of their time with their caregivers. It is very common to find caregivers holding smartphones and they may easily let their children use the smartphones [14–17], mainly to keep them quiet or to prevent them from distracting caregivers during work, to put them to sleep, or to persuade them to eat during mealtimes [16–18, 42]. Caregivers may not recognize the extent or possible effects of the children's smartphone use [16, 36, 42]. A study in Turkey showed that the majority of parents caring for preschool children had installed smartphone apps for their children that were mainly for entertainment and

**Table 3. Potential PSU risk factors.**

| Factors | | B | p | OR | 95% CI |
|---|---|---|---|---|---|
| Child's sex (ref. = male) | Female | 0.416 | .175 | 1.52 | 0.83–2.77 |
| Child's age | | 0.165 | .253 | 1.18 | 0.89–1.57 |
| Family size | | 0.093 | .659 | 1.10 | 0.73–1.66 |
| Parents' employment status (ref. = one of both) | Both | 0.708 | .035 | 2.03 | 1.05–3.91 |
| Household income (ref. = more than 6,000) | 4,000–6,000 | 2.232 | <.001 | 9.32 | 3.07–28.26 |
| | Less than 4,000 | 2.420 | <.001 | 11.24 | 3.56–35.48 |
| Number of preschool children in the household | | 0.145 | .737 | 1.16 | 0.50–2.70 |
| Main caregiver (ref. = grandparent) | Parent | 0.329 | .667 | 1.39 | 0.31–6.21 |
| Main caregiver's age | | 0.058 | .046 | 1.06 | 1.00–1.90 |
| Main caregiver's perception | | 0.226 | .287 | 1.25 | 0.83–1.90 |
| Frequency of smartphone use | | 0.332 | <.001 | 1.39 | 1.32–1.48 |
| Duration of smartphone use (ref. = less than 1 hour) | 1–2 hours | 0.632 | .072 | 1.88 | 0.95–3.75 |
| | More than 2 hours | 2.061 | <.001 | 7.85 | 3.03–20.31 |
| Degree of use | Education | -0.180 | .028 | 0.84 | 0.71–0.98 |
| | Web surfing | -0.076 | .428 | 0.93 | 0.77–1.12 |
| | Games | 0.115 | .103 | 1.12 | 0.98–1.29 |
| | TV/video | 0.160 | .023 | 1.17 | 1.02–1.35 |
| | Music | -0.022 | .775 | 0.98 | 0.84–1.14 |
| | Webtoons/fiction | -0.054 | .658 | 0.95 | 0.75–1.20 |
| | Messenger | 0.145 | .050 | 1.16 | 1.02–1.34 |
| | SNS | 0.016 | .865 | 1.02 | 0.84–1.23 |

Model effect $\chi^2$ = 919.196, p<0.001; Nagelkerke's $R^2$ = 81.2%; Hosmer & Lemeshow $\chi^2$ = 2.531, p = .960; Classification accuracy = 94.6%.

Reference: non-PSU.

Abbreviations: PSU = problematic smartphone use; OR = odds ratio; CI = confidential interval; SNS = social networking services; ref. = reference.

not for education, raising concerns among parents [16]. In one study of parents in six European countries with children between four and six years old, most of parents mentioned that their children like watching TV and most did not worry about their children's TV watching time; for example, there were no planned rules for including an educational aspect [42]. Thus, easy-to-use apps available whenever caregivers need them could be their child's favorite TV shows or videos. Children of preschool age watched TV for an average of half-an-hour to two hours per day for up to a maximum of five hours [16, 42, 43]. In the present study, the average duration of smartphone use in the PSU group was two-and-a-half hours per day, which is about three times more or 100 minutes longer than that of the non-PSU group (0.8 hours per day). TV/video app type was one of the most frequently used apps by children in the PSU group. They may get used to smartphones gradually, similar to TV, and both children and caregivers may be unable to control children's smartphone use [42, 43].

The present study found that the absolute amount of smartphone use time is closely related to PSU and using a smartphone for more than two hours per day substantially increases the risk of PSU (odds ratio 7.85). Previous studies suggested limiting the duration of smartphone use for preschool children to less than half an hour per day to prevent PSU [14, 15]. The American Academy of Pediatrics has recommended limiting total screen time for preschool children to less than one hour per day with high-quality programs [44]. It is therefore necessary to include smartphone use in controlling total screen time for preschool children. First, the caregivers of preschool children should not show smartphones to their children for their

convenience. If they decide to give smartphones to their children, the limits must be clear and well-chosen and consistently enforced so that children will learn to accept the limits and comply with them [20]. The present study additionally found that PSU was associated with parent's employment status, caregiver's age, and household income.

This study has some limitations. First, the data were collected by asking the main caregivers to observe and report on their children's smartphone use at home. Therefore, it is possible that survey results reflect only their viewpoints. Second, this study used a cross-sectional design, which limits it to examining factors associated with PSU among preschool children rather than deriving causal relationships. Third, important factors associated with main caregivers or parents that have not been included in the present study may significantly influence smartphone use. Fourth, children could be exposed to other various screens, but there were not any available data regarding such exposure.

## Conclusions

We found that PSU prevalence among preschool children was similar to that of other age groups, implying a need to pay attention to smartphone use by young children. The most significant use pattern associated with the PSU was the overall duration of smartphone use, indicating that children should keep their smartphone use within one hour per day and balance its function between education and fun for healthy use. Children's smartphone use frequency should be monitored because frequent short-duration smartphone use may be a symptom of PSU. In addition, TV/video app use may be related to the caregivers' tendency to allow TV shows or videos that children like to watch on smartphones without formal rules on limiting screen time. Therefore, to prevent PSU in young children, caregivers should set rules for the total screen time and provide children with more opportunities for physical activities. Furthermore, information should be provided to caregivers on how to set rules for total screen time and to suggest activities to replace children's smartphone use. In addition, to strengthen children's self-regulation of smartphone use, consistent discipline is required.

Smartphone use is continuously increasing, and the age of first smartphone use is gradually decreasing. Countries that have not yet investigated or managed PSU in young children should consider assessing PSU to develop appropriate countermeasures.

## Acknowledgments

Thanks to the institutes and the anonymous participants.

## Author Contributions

**Conceptualization:** Jeong Hye Park.

**Data curation:** Jeong Hye Park, Minjung Park.

**Formal analysis:** Jeong Hye Park.

**Funding acquisition:** Minjung Park.

**Investigation:** Jeong Hye Park, Minjung Park.

**Methodology:** Jeong Hye Park.

**Project administration:** Jeong Hye Park.

**Supervision:** Jeong Hye Park.

**Validation:** Jeong Hye Park, Minjung Park.

**Visualization:** Jeong Hye Park.

**Writing – original draft:** Jeong Hye Park, Minjung Park.

**Writing – review & editing:** Jeong Hye Park, Minjung Park.

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
