## [Decision Letter · Decision Letter 0]

23 Jul 2020

PONE-D-20-05258

Smartphone use patterns and problematic smartphone use among preschool-age children

PLOS ONE

Dear Dr. Park,

Thank you for submitting your manuscript to PLOS ONE. After careful consideration, we feel that it has merit but does not fully meet PLOS ONE’s publication criteria as it currently stands. Therefore, we invite you to submit a revised version of the manuscript that addresses the points raised during the review process.

We look forward to receiving your revised manuscript.

Kind regards,

Tzipi Horowitz-Kraus, PhD

Academic Editor

PLOS ONE

Journal Requirements:

3. We note you have included a table to which you do not refer in the text of your manuscript. Please ensure that you refer to Table 2 in your text; if accepted, production will need this reference to link the reader to the Table.

Additional Editor Comments (if provided):

Reviewers' comments:

Reviewer's Responses to Questions

**Comments to the Author**

1. Is the manuscript technically sound, and do the data support the conclusions?

Reviewer #1: Partly

Reviewer #2: Partly

2. Has the statistical analysis been performed appropriately and rigorously? 

Reviewer #1: Yes

Reviewer #2: Yes

3. Have the authors made all data underlying the findings in their manuscript fully available?

Reviewer #1: Yes

Reviewer #2: Yes

4. Is the manuscript presented in an intelligible fashion and written in standard English?

Reviewer #1: No

Reviewer #2: Yes

5. Review Comments to the Author

Reviewer #1: *Did all children in this age group go to daycare centers or kindergartens?; was there anyone at homecare? It must be stated in the text.

*Smartphone usage time; How was it calculated? what about weekdays and weekend usage times?

*Did children always use the smartphone with their parents?

*Exposure to other screens should be stated as a confounding factor, if no data was avaliable, it should be stated as a limitation of the study.

*Parental ages and education levels, and number of sibling should be given.

*English editing is necessary.

Reviewer #2: This manuscript describes phone surveys of caregivers with children ages 3-6 years (N= 1,378) to examine the relation between smartphone use patterns and problematic smartphone use (PSU) in preschoolers in South Korea. Based on a nationally representative sample, the authors found that 17% of children were classified as a PSU group. Also, the likelihood of PSU was positively predicted by frequent smartphone use, more than 2 hours of daily media use, and higher exposure to TV shows and videos. Also, the author found that family characteristics (caregiver age, parent employment status, household income) were associated with PSU. The authors concluded that problematic smartphone use is prevalent among preschoolers and emphasized the role of caregivers in preventing PSU among preschoolers.

The problematic smartphone use among young children is a relatively understudied topic in the field. This manuscript fills in this gap by describing preschoolers’ problematic smartphone usage and exploring demographic and media use factors. Also, the author presented a representative sample with a good reflection of the variations and diversity, enhancing the generalizability of the results.

Having said that, I think further information is required in the introduction, methods, and discussion sections to understand and interpret the findings clearly. First, in the introduction section, I think the author should provide more details on theoretical or conceptual justifications for studying problematic smartphone use in preschoolers by connecting their research question with the existing literature on predictors and outcomes of screen time in preschoolers. Next, the authors should clarify their research procedure in the method section. The design and analysis seem appropriate, but further details are necessary. In particular, it was not clear to me how the authors measured the frequency and amount of smartphone use. Also, I would like to see how the ‘education’ category was defined and how it would differ from the rest of the categories. Lastly, the discussion section requires a revision. In the discussion, some of the interpretations are broad statements that seem to go beyond data. Also, the discussion section includes is a very long paragraph that contains several different ideas, which should be restructured and revised to improve clarity and flow. Here are my specific questions, comments, and suggestions.

Introduction

Page 4. The authors stated that “PSU in preschool-age children has not yet been well studied” in the introduction. I think it is a valid point, but this could be connected to the existing literature on predictors and outcomes of screen time in preschoolers (see Tandon et al.., 2011; Tamana et al., 2019). That way, the authors will have a foundation to tie their research to the field of young children and media effects throughout the manuscript (explaining their rationales, interpreting the results, and highlighting their contributions).

Tandon, P. S., Zhou, C., Lozano, P., & Christakis, D. A. (2011). Preschoolers’ total daily screen time at home and by type of child care. The Journal of pediatrics, 158(2), 297-300.

Tamana, S. K., Ezeugwu, V., Chikuma, J., Lefebvre, D. L., Azad, M. B., Moraes, T. J., ... & Dick, B. D. (2019). Screen-time is associated with inattention problems in preschoolers: Results from the CHILD birth cohort study. PloS one, 14(4), e0213995.

Methods

Page 6. The authors included “education” as one of the app types. However, it was not clear to me how the "education" category was defined. I was wondering how education would differ from the rest of the categories. Children may be playing games or watching educational television programs. There is a large body of research showing the importance of media content and not just the amount of screen time within this age group. For example, a longitudinal study showed a positive impact of educational media such as Sesame Street on language and cognitive development in young children. On the other hand, entertainment content has shown to have no or negative effect, and violent media content has negative consequences (see Kirkorian et al., 2008).

Kirkorian, H. L., Wartella, E. A., & Anderson, D. R. (2008). Media and young children's learning. The Future of children, 39-61.

Page 6. Please provide sufficient details on how smartphone use patterns (frequency, duration) were measured. In its current form, it is hard for another researcher to reproduce the survey items described. I am not sure if the recall was based on a “typical” day method (i.e., estimate the amount of screen time in a “typical” day) or a diary method (e.g., thinking of yesterday, how much time did you spend..; see Vandewater & Lee, 2009).

Vandewater, E. A., & Lee, S. J. (2009). Measuring children's media use in the digital age: issues and challenges. American Behavioral Scientist, 52(8), 1152-1176.

Results

Page 7. “The study sample comprised 201 parents of children under 10 years old.” I had to pause and go back to double-check whether this refers to the sample in the current study or whether it represents the sample for the prior research on scale development. I think it is the former, but it would be helpful to state the relevant study clearly.

Pages 9-10: In the results section, the authors used “participants” to describe children. Here are some examples: 1) the first subheading “Differences between non-PSU and PSU by participant and family characteristics,” 2) Table 1 title “Differences between non-PSU and PSU according to participant and family characteristics” and 3) Table 1 under Family characteristics “Number of participants in the household.” Given that this study is based on parent reports, labeling children as participants may lead to some confusion. The authors may consider replacing “participants” with “children” in the results section for clarity.

Pages 10. As I mentioned earlier in the introduction section, the “education” category is not clear. How does it differ from games and TV/video?

Discussion

Page 14. The authors said, “Watching TV/videos is a well-established cause of increased screen time and sedentary behavior [12, 34-36].” I am not sure this statement is supported by the sources cited. If I understood correctly, the cited sources mainly focus on associations rather than providing any “causal” evidence.

Pages 14-15. The paragraph starting with “Among the potential risk factors” is very long and contains several different points, which I found hard to follow. This should be split into two or more paragraphs and revised to increase clarity.

Page 15. In the discussion, the author listed the findings from different prior studies. I think these previous studies need to be described with a bit more details about their samples so that readers can understand the contexts of their findings: “Caregivers mentioned that their children like watching TV and most did not worry about their children’s TV watching time; for example, there were no planned rules for including an educational aspect [22]. Children of this age watched TV for an average of 0.5 - 2 hours per day for up to a maximum of five hours [6, 22, 38].”

Page 16. I am not sure if the following statement is accurate based on the data presented in the manuscript: “Our findings that PSU decreased when one parent worked, but not when both worked, and that PSU increased with older main caregivers suggests that while one parent is at work, the other parent, not a grandparent, is caring for the child all day.” That is, how does this finding suggest that the other parent is caring for the child all day? I think this should be replaced with the authors’ interpretation of the results (i.e.., PSU was associated with parent’s employment status and caregiver’s age).

Page 16. “We also found that PSU was less prevalent in higher-income households, suggesting that more attention and care provided by the main caregiver allows children to avoid excessive smartphone use.” The authors made a direct connection between household income and the level of attention and care children received from their caregivers. I think this is a very broad claim that seemed to go beyond data. I suggest the authors reconsider this.

Minor points

Page 22. 33. There is a typo: preschoolers inn childcare -> preschoolers in childcare

6. PLOS authors have the option to publish the peer review history of their article (what does this mean?). If published, this will include your full peer review and any attached files.

Reviewer #1: No

Reviewer #2: No

---

## [Author Response · Author response to Decision Letter 0]

21 Sep 2020

We thank you and the reviewers for your thoughtful suggestions and insights, which have enriched the manuscript and helped us to produce a more concise and higher quality account of our research.

We attached the file 'Response to Reviewers'.

---

## [Editor Report · Decision Letter 1]

8 Dec 2020

Smartphone use patterns and problematic smartphone use among preschool children

PONE-D-20-05258R1

Dear Dr. Park,

We’re pleased to inform you that your manuscript has been judged scientifically suitable for publication and will be formally accepted for publication once it meets all outstanding technical requirements.

Kind regards,

Tzipi Horowitz-Kraus, PhD

Academic Editor

PLOS ONE
---

## [Editor Report · Acceptance letter]

2 Feb 2021

PONE-D-20-05258R1 

Smartphone use patterns and problematic smartphone use among preschool children 

Dear Dr. Park:

I'm pleased to inform you that your manuscript has been deemed suitable for publication in PLOS ONE. Congratulations! Your manuscript is now with our production department. 

Kind regards, 

on behalf of

Dr. Tzipi Horowitz-Kraus 

Academic Editor

PLOS ONE